# Nearest-neighbour resonating valence bonds in $YbMgGaO_4$

Yuesheng Li[1,2], Devashibhai Adroja[3,4], David Voneshen[3], Robert I. Bewley[3], Qingming Zhang[2,5,6], Alexander A. Tsirlin[1] & Philipp Gegenwart[1]

Since its proposal by Anderson, resonating valence bonds (RVB) formed by a superposition of fluctuating singlet pairs have been a paradigmatic concept in understanding quantum spin liquids. Here, we show that excitations related to singlet breaking on nearest-neighbour bonds describe the high-energy part of the excitation spectrum in $YbMgGaO_4$, the effective spin-1/2 frustrated antiferromagnet on the triangular lattice, as originally considered by Anderson. By a thorough single-crystal inelastic neutron scattering study, we demonstrate that nearest-neighbour RVB excitations account for the bulk of the spectral weight above 0.5 meV. This renders $YbMgGaO_4$ the first experimental system where putative RVB correlations restricted to nearest neighbours are observed, and poses a fundamental question of how complex interactions on the triangular lattice conspire to form this unique many-body state.

[1] Experimental Physics VI, Center for Electronic Correlations and Magnetism, University of Augsburg, 86159 Augsburg, Germany. [2] Department of Physics, Renmin University of China, Beijing 100872, China. [3] ISIS Pulsed Neutron and Muon Source, STFC Rutherford Appleton Laboratory, Harwell Campus, Didcot, Oxfordshire OX11 0QX, UK. [4] Highly Correlated Matter Research Group, Physics Department, University of Johannesburg, PO Box 524, Auckland Park 2006, South Africa. [5] Department of Physics and Astronomy, Shanghai Jiao Tong University, Shanghai 200240, China. [6] Collaborative Innovation Center of Advanced Microstructures, Nanjing 210093, China. Correspondence and requests for materials should be addressed to Y.L. (email: yuesheng.man.li@gmail.com) or to Q.Z. (email: qmzhang@ruc.edu.cn).

Quantum spin liquid (QSL) is a long-sought exotic phase in condensed matter physics. It is intimately related to the problem of high-temperature superconductivity and may be instrumental in realizing topological quantum computation[1–6]. In a QSL, spins are highly entangled up to long distances and times without symmetry breaking down to zero temperature due to strong quantum fluctuations[3]. Experimental systems exhibiting QSL behaviour are actively sought after. However, most of the existing materials are suffering from magnetic defects[7,8], spatial coupling anisotropy[8–10] and (or) antisymmetric Dzyaloshinsky–Moriya anisotropy[11]. Recently, a triangular QSL candidate YbMgGaO$_4$ attracted much interest[12–15], because it seems to be free from all of the above effects. Neither spin freezing nor long-range ordering were detected by muon spin relaxation (μSR) down to 0.048 K (ref. 14). Together with the absence of any residual spin entropy[12], this renders YbMgGaO$_4$ a unique material that may exhibit a gapless U(1) QSL ground state.

A QSL state can be represented by a superposition of many different partitions of a system into valence bonds (spin-0 singlet pairs)[3], as proposed by Anderson back in 1973 (refs 1,2). Such valence bonds can be formed between nearest-neighbour spins and between spins beyond nearest neighbours. The longer the bond, the weaker the respective singlet pairing energy. Low-energy excitations arise from breaking long-range valence bonds or rearranging the short bonds into longer ones[3,16]. High-energy excitations result from breaking nearest-neighbour valence bonds. Therefore, for characterizing a QSL, the detailed investigation of both high- and low-energy excitations is required.

In YbMgGaO$_4$, excellent transparence with the optical gap exceeding ~3 eV and the robust insulating behaviour with the unmeasurably high resistance suggest a large charge gap, placing the material deep in the Mott-insulator regime of the Hubbard model. Strong localization of the 4$f$ electrons of Yb$^{3+}$ should restrict magnetic interactions to nearest neighbours ($S_1$ and $S_2$), but these interactions are anisotropic[13],

$$
\begin{aligned}
\mathbf{H} = {} & J_{zz} S_1^z S_2^z + J_{\pm} \left( S_1^+ S_2^- + S_1^- S_2^+ \right) \\
& + J_{\pm\pm} \left( \gamma_{12} S_1^+ S_2^+ + \gamma_{12}^* S_1^- S_2^- \right) \\
& - \frac{i J_{z\pm}}{2} \left( \gamma_{12}^* S_1^+ S_2^z - \gamma_{12} S_1^- S_2^z + \langle 1 \leftrightarrow 2 \rangle \right),
\end{aligned}
\tag{1}
$$

owing to the strong spin-orbit coupling, where the local moment $S = 1/2$ is a pseudospin, that is, a combination of spin and orbital moments[15,17–19]. The lowest-energy eigenstate of a dimer formed by such anisotropic pseudospins is, nevertheless, a pure singlet, $(1/\sqrt{2})(|\uparrow\downarrow\rangle - |\downarrow\uparrow\rangle)$, with the energy $-3/4 J_0$ for the antiferromagnetic isotropic coupling, $J_0 \equiv (4J_{\pm} + J_{zz})/3 = 0.13(1)$ meV (ref. 13), as observed experimentally. In contrast to Heisenberg spins, the Yb$^{3+}$ pseudospins do not form a three-fold degenerate triplet state and feature three non-degenerate excited states separated by $0.809 J_0$, $1.012 J_0$ and $1.179 J_0$ from the singlet state instead. Excitations of a system can be viewed as the transitions between the singlet ground state and one of the excited states. Therefore, the resonating valence bond (RVB) picture holds, albeit with minor quantitative modifications due to the different structure of the excited states.

Two very recent inelastic neutron scattering (INS) studies reported a continuum of spin excitations in YbMgGaO$_4$ in the energy range between 0.25 and 1.5 meV (refs 20,21), and a phenomenological interpretation of these excitations in terms of a spinon Fermi surface has been proposed[20]. However, given the nearest-neighbour magnetic energy of $J_0 = 0.13(1)$ meV only[13], the excitations were observed at energies between $2J_0$ and $10J_0$. Therefore, they are high-energy magnetic excitations of YbMgGaO$_4$.

In this paper, we propose a different interpretation of these high-energy excitations and also endeavour to probe YbMgGaO$_4$ at lower energies. This task is extremely challenging, owing to the low energy scale of $J_0$ and the limits of instrumental energy resolution for neutron spectrometers. We report a thorough INS investigation of a single crystal of YbMgGaO$_4$ at energies between 0.02 and 3.5 meV, that is, 0.15–27 in units of $J_0$. We present the data collected at the low temperature of 0.1 K, which is well inside the gapless ground-state regime defined by the saturation of the μSR rate[14], and at a much higher temperature of 35 K corresponding to $23 J_0$. The high-energy excitations observed previously[20,21] are confirmed and ascribed to nearest-neighbour RVB correlations. At low temperatures, these excitations are suppressed at energies below $J_0$, which suggests their gapped nature. Our results imply that distinct gapless excitations should exist at much lower energies, and we indeed observe traces of such excitations at the lowest energies accessible in our experiment.

## Results

**High energy nearest-neighbour RVB correlations.** The INS data for YbMgGaO$_4$ are shown in Figs 1 and 2. A continuum of excitations broadly distributed in both momentum (**Q**) (see Fig. 1) and energy ($0.1 \leq \hbar\omega \leq 2$ meV) space (see Fig. 2) is clearly visible. At 0.1 K, external field shifts the spectral weight towards higher energies (see Fig. 2), thus indicating the magnetic origin of these excitations. Remarkably, the excitation continuum persists up to 35 K, that is, at a temperature that is 23 times higher than $J_0$. In fact, there are no qualitative differences between the high-energy parts of the INS spectra measured at 0.1 and 35 K apart from a 2.57(4)-fold reduction in the intensity near the hump centre ~0.7 meV (see Fig. 2) when the temperature is increased to 35 K. The wave-vector and temperature dependence of the excitation continuum clearly indicates its spin–spin correlation origin and excludes other possible interpretations, such as CEF excitations, which are **Q**-independent and observed at energies larger than 39 meV (refs 13,15,21).

We first focus on the wave vector dependence of the INS intensity measured with the incident neutron energy of $E_i = 5.5$ meV. Assuming uncorrelated nearest-neighbour valence bonds on a triangular lattice, the equal-time INS intensity can be expressed as ref. 22

$$
N|F(\mathbf{Q})|^2 = \frac{2}{3} N|f(\mathbf{Q})|^2 \{3 - \cos(2\pi H) - \cos(2\pi K) - \cos[2\pi(H+K)]\}.
\tag{2}
$$

Here, $f(\mathbf{Q})$ is the magnetic form factor of free Yb$^{3+}$, and $N$ is the total number of nearest-neighbour valence bonds probed in the INS measurement. This expression accounts for the experimental spectral weight above 0.5 meV, thus suggesting that at high energies spin–spin correlations are restricted to nearest neighbours. Any static state, such as valence bond solid[23] and glass[24,25], is excluded by our previous μSR study[14], and the RVB scenario turns out to be most plausible, as supported by the following arguments:

First, the **Q**-dependence of the INS signal at 0.1 and 35 K (after the subtraction of the background term $b$) is well described by the uncorrelated nearest-neighbour valence bond model on a triangular lattice (see Fig. 1c–f). No signatures of spin–spin correlations beyond nearest neighbours are observed (Supplementary Note 2 and Supplementary Figs 10 and 11). This **Q**-dependence cannot be understood by short distance correlations in an arbitrary ground state on the triangular lattice. For example, the 120° long-range order would produce spin-wave excitations[26] and a qualitatively different **Q**-dependence even at

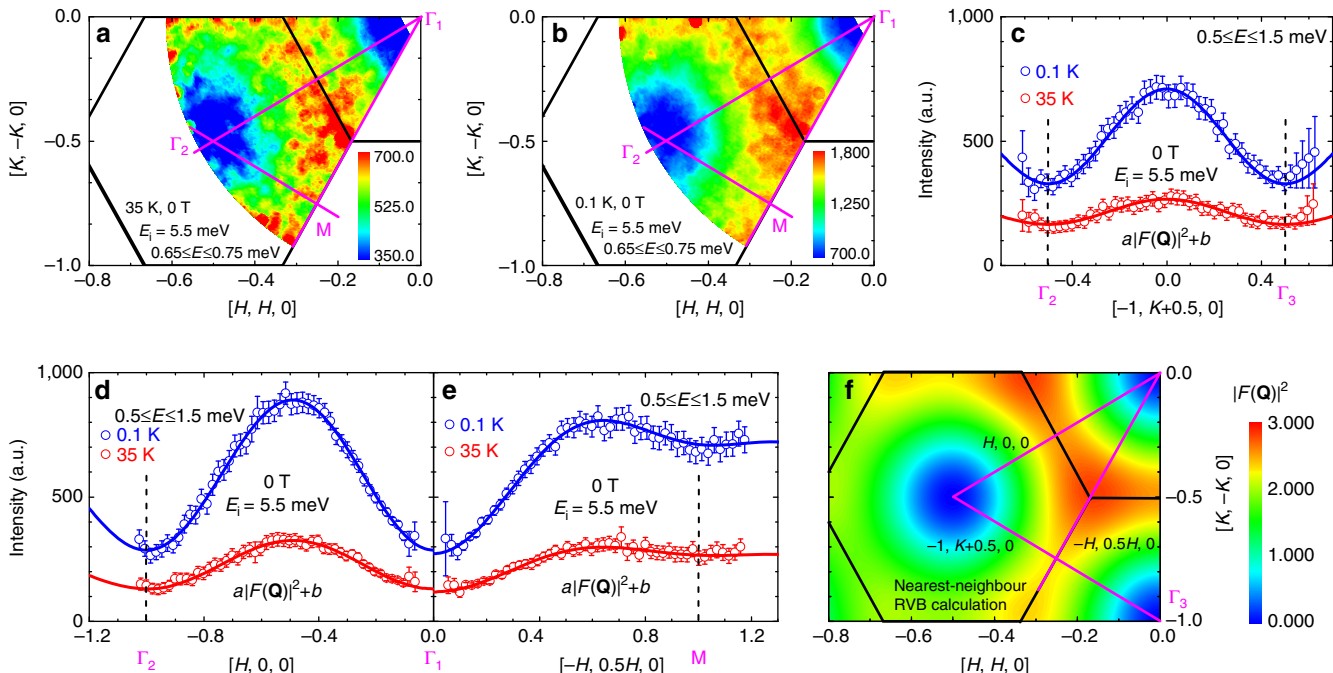

**Figure 1 | Wave-vector dependences of the INS intensity for YbMgGaO$_4$.** Wave-vector dependences of excitations measured under 0 T at 35 K (**a**) and at 0.1 K (**b**). Wave-vector dependences of the INS intensity along [ − 1, K + 0.5, 0] (**c**), [H, 0, 0] (**d**), and [ − H, 0.5H, 0] (**e**), with lines representing the calculated nearest-neighbour RVB dependence. Error bars on INS data indicate one standard error propagated from neutron counts (using Horace-Matlab). Calculated $|F(\mathbf{Q})|^2$ (from equation (2)) (**f**). The black lines represent Brillouin zone boundaries. Pink lines show the high-symmetry directions with special reciprocal-space points labelled. Note that the experimental data contain a **Q**-independent background, which is about the same at $\Gamma_1$, $\Gamma_2$ and $\Gamma_3$. This background is missing in the RVB calculation in **f**, where $I_\Gamma = 0$.

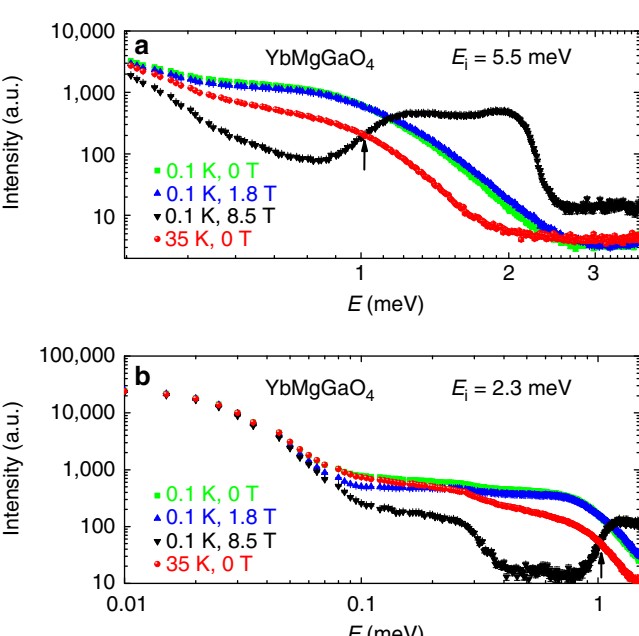

**Figure 2 | Energy dependence of the INS intensity for YbMgGaO$_4$.** The excitation continuum was probed with the incident neutron energy of 5.5 meV (**a**) and 2.3 meV (**b**). Black arrows show the lower boundary (gap energy) of the spin-wave excitations in the fully polarized state. This boundary is determined as the energy, where the high-temperature (35 K, 0 T) INS intensity crosses the low-temperature (0.1 K, 8.5 T) one. Error bars on INS data indicate one standard error propagated from neutron counts (using Horace-Matlab), and all measured **Q** space is integrated.

high energies (Supplementary Note 5 and Supplementary Figs 21–26).

Second, the antiferromagnetic nature of the isotropic nearest-neighbour coupling, $J_0 \equiv (4J_\pm + J_{zz})/3 = 0.13(1)$ meV (ref. 13), allows the formation of spin singlet in a pair of the Yb$^{3+}$ spins (Supplementary Note 1 and Supplementary Fig. 1).

Third, temperature dependence of the pre-factor $a$ in the RVB expression, $a(35\,\text{K})/a(0.1\,\text{K}) \sim 0.3$ (Supplementary Table 1), is consistent with the expected ratio,

$$\frac{N(T_2 = 35\,\text{K})}{N(T_1 = 0.1\,\text{K})} \sim \frac{1 + \exp\left(-\frac{0.809 J_0}{k_B T_1}\right) + \exp\left(-\frac{1.012 J_0}{k_B T_1}\right) + \exp\left(-\frac{1.179 J_0}{k_B T_1}\right)}{1 + \exp\left(-\frac{0.809 J_0}{k_B T_2}\right) + \exp\left(-\frac{1.012 J_0}{k_B T_2}\right) + \exp\left(-\frac{1.179 J_0}{k_B T_2}\right)} \sim 0.26,$$

(3)

based on the thermal distribution of the eigenstates of the Yb$^{3+}$ dimer. With increasing temperature, a larger fraction of nearest-neighbour singlets is excited.

Fourth, the uniform spin susceptibility, $\chi'(E)$, which is obtained from the INS spectrum measured around the Gamma point (**Q** = 0) via the fluctuation-dissipation theorem and the Kramers − Kronig transformation[22], is almost zero at 0.1 K above ∼0.5 meV, in agreement with the proposed RVB state (Supplementary Note 3 and Supplementary Figs 12 and 13).

Fifth, the energy dependence of the integrated INS signal reveals gapped nature of the high-energy excitations (see below for the details), which is consistent with the aforementioned suppression of the uniform susceptibility above ∼0.5 meV.

Last, both spin and valence bond freezing are excluded by our µSR measurement reported previously[14].

The above six arguments suggest that the whole excitation continuum at energies above $J_0$ may be due to the nearest-neighbour RVB-type correlations. We prove this explicitly above 0.5 meV, while below 0.5 meV the **Q**-dependent

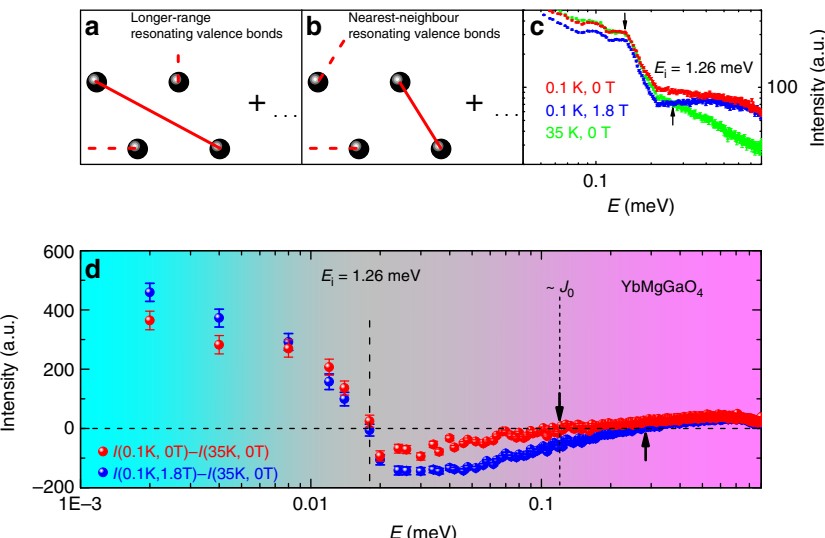

**Figure 3 | INS data for YbMgGaO$_4$ with the incident energy of 1.26 meV.** (**a**,**b**) Sketch of longer-range and nearest-neighbour RVBs on the triangular lattice (for one unit cell). The red lines represent the valence bonds (spin-singlets). (**c**) Raw INS spectra (**d**) integrated INS intensities at 0.1 K under 0 and 1.8 T, after the subtraction by the corresponding spectrum measured at 35 K (0 T). Black arrows show the lower boundaries of the continuum excitations. The dashed vertical line indicates the low-energy crossing of the intensities. Error bars on INS data indicate one standard error propagated from neutron counts (using Horace-Matlab), and all measured **Q** space is integrated in **c**,**d**.

data measured with the incident energy $E_i = 5.5$ meV are contaminated by the elastic signal (Supplementary Figs 14–19). Lower energies can be probed with $E_i = 1.26$ meV (the energy resolution $\sigma \sim 20\,\mu$eV (ref. 27)), but these data cover a limited **Q**-range only. Nevertheless, we find no qualitative differences between the spectra at $\sim 0.3$ and $\sim 0.7$ meV in all measured **Q** space ($E_i = 1.26$ meV) apart from an overall increase in the intensity. This indicates same, nearest-neighbour nature of spin–spin correlations across the whole excitation continuum above $J_0$ that was previously ascribed to the spinon Fermi surface.

It is crucial, though, that this continuum and the associated nearest-neighbour spin–spin correlations do not persist down to zero energy, because the nearest-neighbour RVBs are gapped, whereas YbMgGaO$_4$ clearly shows gapless behaviour[12,14]. Therefore, the RVB scenario holds at high energies only. The presence of a distinct low-energy regime is supported by the analysis of the energy-dependent spectra integrated over all measured **Q** space.

**Low energy long-range spin correlations**. For energy transfer below $J_0$, excitations related to the breaking of nearest-neighbour spin singlets must freeze out as long as thermal energy is insufficient to overcome $J_0$, that is, $T < 1.5$ K. We, therefore, expect that below 0.13 meV the INS intensity at 0.1 K falls below that at 35 K. As indicated by the downward-pointing arrow in Fig. 3c, this expected crossing of the overall scattering intensity is observed indeed. Respectively, the intensity difference $I(0.1\,\mathrm{K}) - I(35\,\mathrm{K})$ at zero magnetic field changes sign and becomes negative at energy transfer below $J_0$ (Fig. 3d).

Further information is obtained from the INS spectra at finite magnetic fields applied along the crystallographic $c$-direction. At 8.5 T, which fully polarizes the moments at low temperatures[13,15], a clear boundary is observed in the low-energy magnetic excitations, leading to a crossing of $I(0.1\,\mathrm{K}, 8.5\,\mathrm{T})$ with $I(35\,\mathrm{K})$ near 1 meV, as indicated by the arrow in Fig. 2. This gap is related to the Zeeman energy[15,21] in the applied field of 8.5 T. In the same vein, under a moderate applied field of 1.8 T, which polarizes the spins only partially, negative values of $I(0.1\,\mathrm{K}) - I(35\,\mathrm{K})$ occur below 0.27 meV (see Fig. 3d). This energy

lies in between $J_0$ and the Zeeman energy $\mu_0\mu_\mathrm{B}g_\parallel H_\parallel = 0.39$ meV of spin-wave excitations for this field. When the field is reduced to zero, the crossing of intensities shifts to $J_0$ (Fig. 3c). We, therefore, associate this effect with an energy gap for the continuum of nearest-neighbour RVB-type excitations[3]. These excitations seem to be unrelated to the gapless spinon Fermi surface, in contrast to recent expectations based on the INS measurements at higher energies[20].

It is worth noting that a qualitatively similar crossing of the INS intensities measured at low and high temperatures has been recently observed in the frustrated pyrochlore Er$_2$Ti$_2$O$_7$ (ref. 28), where magnetic excitations are gapped. In our case, the relation $I(0.1\,\mathrm{K}, 0\,\mathrm{T}) < I(35\,\mathrm{K}, 0\,\mathrm{T})$ is also clearly detected in zero field at transfer energies from 0.13 meV down to 0.018 meV, below which a rapid increase of the low-temperature intensity sets in. This lower energy is roughly the same as the energy resolution $\sigma \sim 20\,\mu$eV ($0.15J_0$) (ref. 27) of the LET spectrometer at the incident neutron energy of 1.26 meV. We emphasize that the inelastic signal does not become featureless at this energy ($\sigma$), as otherwise a smooth convoluted Lorentzian-Gaussian peak profile would be expected (see the raw data in Fig. 3c and Supplementary Fig. 20). At low transfer energies, the inelastic signal is found on top of the elastic background (see Fig. 3c). Assuming a weakly temperature-dependent elastic signal at $T \leq 35$ K, we expect that it cancels out when analysing $I(0.1\,\mathrm{K}) - I(35\,\mathrm{K})$. Therefore, the intensity difference observed in zero field (see Fig. 3d) is intrinsic, as further confirmed by its tangible field dependence, and should reflect the onset of low-energy excitations related to longer-range correlations in YbMgGaO$_4$ (refs 12,14). The most conspicuous effect of this change is the shift of the intensity maxima from the K-points in the high-energy regime to the M-points in the low-energy regime (Supplementary Note 4 and Supplementary Fig. 18), as also seen in the diffuse scattering reported by Paddison et al.[21]

**Discussion**

The clear separation between the low-[12,14] and high-energy excitations in the spectrum of YbMgGaO$_4$ (see Fig. 3d) is interesting and unique, rendering YbMgGaO$_4$ distinct from QSL

materials known to date, such as herbertsmithite[7,29], organic charge transfer salts[9,10] and $Ca_{10}Cr_7O_{28}$ reported recently[30]. The RVB scenario on the triangular-lattice was also discussed for the cluster magnet $LiZn_2Mo_3O_8$, where a spin-liquid state with both nearest-neighbour and next-nearest-neighbour correlations is formed[31–33]. It is also worth noting that the continuum of nearest-neighbour RVB excitations goes back to the original idea by Anderson[1] who argued that Heisenberg spins on the regular triangular lattice evade long-range magnetic order and form the nearest-neighbour RVB QSL state. Although Anderson's conjecture was not confirmed in later studies[34], the formation of a QSL on a triangular lattice with spatial anisotropy[35], next-nearest-neighbour couplings[36] and multiple-spin exchange[37] was identified in the recent literature. Whereas the multiple-spin exchange can be clearly excluded due to the strongly localized nature of the $4f$ electrons of $Yb^{3+}$, two other effects are potentially relevant to $YbMgGaO_4$.

The presence of next-nearest-neighbour couplings is currently debated based on the modelling of the magnetic diffuse scattering[21,38]. Spatial anisotropy of nearest-neighbour couplings can be, at first glance, excluded, based on the three-fold symmetry of the crystal structure[12]. However, recent experiments[15,21], including our INS study[15] of crystal-field excitations of $Yb^{3+}$, pinpoint the importance of the Mg/Ga disorder that leads to variations in the local environment of $Yb^{3+}$. An immediate effect of this structural disorder is the distribution of g-values that manifests itself in the broadening of excitations in the fully polarized state, yet randomness of magnetic couplings resulting in local spatial anisotropy seems to be relevant too[15,18,21].

Our result suggests that the broad excitation continuum in $YbMgGaO_4$ reflects nearest-neighbour spin correlations and bears no obvious relation to the gapless spinon Fermi surface, a conclusion consistent with the absence of the Fermi spinon or any other magnetic contribution to the thermal conductivity[39]. On the other hand, gapless nature of $YbMgGaO_4$ evidenced by the non-zero low-temperature susceptibility[12,14] and the power-law behaviour of the magnetic specific heat[12] are indicative of a distinct low-energy regime that has been glimpsed in our experiment. These low-energy excitations are likely to contain crucial information on whether the ground state of $YbMgGaO_4$ is indeed a QSL, or a special case of the disorder-induced mimicry of a spin liquid, as proposed recently[40,41].

## Methods

**Sample preparation.** Large single crystals ($\sim 1$ cm) of $YbMgGaO_4$ were grown by the floating zone technique reported previously[13]. The as grown rod ($\sim 50$ g) was cut into slices along the *ab*-plane (the easily cleavable direction). Ten best-quality *ab*-slices of the single-crystal (total mass $\sim 10$ g) were selected for the neutron scattering experiment on LET by Laue X-ray diffractions on all surface (Supplementary Figs 2 and 3). The slices were fixed to the copper base by Cytop glue to avoid any shift in an applied magnetic field up to 8.5 T.

**Neutron scattering measurements.** Systematic neutron scattering experiments were carried out on a cold neutron multi-chopper spectrometer LET at the ISIS pulsed neutron and muon source. Incident energies of 26.8, 5.5, 2.3 and 1.26 meV were chosen for both elastic and inelastic scattering with the energy resolution of 1,400, 160, 48 and 20 μeV, respectively[27]. The sample temperature of 0.1 K was achieved using dilution refrigerator. The neutron diffraction (elastic signal) showed that the alignment of the single crystals was sufficient for the INS study of the continuous excitations. No additional diffraction peaks were observed down to 0.1 K, compatible with the absence of long-range magnetic order (Supplementary Figs 4–6). All neutron scattering data were processed and analysed using Horace-Matlab[42] on the ISIS computers. Asymmetry of the intensities was observed due to the macro-scale non-rotational symmetry of the sample around the rotation axis. For the sake of clarity, the raw data have been symmetrized and averaged using the point symmetry ($D_{3d}$) in the reciprocal lattice space (see Fig. 1a,b). The corresponding raw data can be found in Supplementary Figs 7–9.

External magnetic fields of 1.8 and 8.5 T were applied along the *c*-axis. The data sets in Fig. 1a,b were integrated over the momentum space, $-0.9 \le \eta \le 0.9$ in $[0, 0, -\eta]$, and over a small energy range, $0.65 \le E \le 0.75$ meV. The data sets in Fig. 1c–e were integrated over the momentum space, $-1.03 \le \xi \le -0.97$ in $[\xi, -\xi/2, 0]$, $-0.03 \le \xi \le 0.03$ in $[\xi/2, -\xi, 0]$, and $-0.03 \le \xi \le 0.03$ in $[0, \xi, 0]$, respectively. All data sets in Fig. 1c–e were integrated over the same momentum range, $-0.9 \le \eta \le 0.9$ in $[0, 0, -\eta]$, and over the same energy range, $0.5 \le E \le 1.5$ meV. *a* and *b* are fitted constants for the proportionality and background, respectively (see Fig. 1c–e and Supplementary Table 1). The data sets in Figs 2 and 3 were integrated over all measured momentum space.

**Data availability.** The data sets generated during and/or analysed during the current study are available from the corresponding author on reasonable request.

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

## Acknowledgements

We thank Gang Chen, Haijun Liao, Changle Liu, Sasha Chernyshev and Mike Zhitomirsky for helpful discussions. Y.L. was supported by the start-up funds of Renmin University of China. Q.Z. was supported by the Fundamental Research Funds for the Central Universities, and by the Research Funds of Renmin University of China. This work was supported by the NSF of China (No. 11474357) and the Ministry of Science and Technology of China (973 Project No. 2016YFA0300504). The work in Augsburg was supported by the German Science Foundation through TRR-80 and the German Federal Ministry for Education and Research through the Sofja Kovalevskaya Award of the Alexander von Humboldt Foundation.

## Author contributions

Y.L., D.A. and Q.Z. planned the experiments. Y.L. synthesized and characterized the sample. Y.L., D.V., R.I.B. and Q.Z. collected the neutron scattering data. Y.L. analysed the data. Y.L., A.A.T. and P.G. wrote the manuscript with comments from all co-authors. The manuscript reflects the contributions of all authors.

## Additional information

**Competing interests:** The authors declare no competing financial interests.

