## [Peer Review File · Nature Communications]

Reviewers' comments:

Reviewer #1 (Remarks to the Author):

In this work the authors present a new interpretation of the data of INS based on more accurate and better resolved experiments in YbMgGaO_4 , a material that according to the authors should represent a rather clean realization of the isotropic Heisenberg model on the triangular lattice.

The results may be interesting, but in my opinion the interpretation of the data is not as clear as the authors claim.

The authors state that Eq.1 univocally identifies the INS spectrum of a short range RVB. However I have several doubts.

Any smooth function of the momenta,

satisfying the spatial symmetry of the triangular lattice and

that vanishes at the Gamma point (as a consequence of spin conservation at $T=0$) can be approximated by an expression like that (just because higher harmonics are much weaker in a smooth function expected at high T).

At high temperature this is

certainly the fingerprint that in this material the spins lie in a triangular lattice (apart of the fact that this function

does no longer need to vanish at Gamma),

but nothing more than that.

The surprising fact is that the authors find that this ansatz for the INS is essentially valid also at very low temperatures (0.1K about 150 times smaller than the expected J_0). This is certainly interesting but I found the data hard to believe and to understand (e.g. the momenta notations) at low temperatures.

First let me discuss about the notations. In Eq.1 the signal vanishes correctly for $H=K=0$ but not for $H,K=-0.5$ as is instead reported in Fig.1f.

It looks to me that Eq.1 is not typed well and should be

substituted by the one reported in the supplementary information (Eq.S2).

Second, assuming the above is correct the experimental data are also hard to believe.

The panels a and b are qualitatively different apart for colors from the panel f. At Gamma there is a huge signal both in a and b.

The fact that they are

colored in the same way of panel f is a bit confusing.

As in the ansatz, at $T=0$ the signal should be 0 at Gamma if we are dealing with a spin liquid, namely at $(K,H) = (0,0)$ or $(-0.5,-0.5)$ in their way to represent the BZ.

At finite temperature, if the spin susceptibility is assumed to be at most

finite at $T=0$ the signal at Gamma should go down to zero at least as T , whereas the signal at Gamma increases at $T=0.1\text{K}$ (!) as compared to the $T=35\text{K}$. The data look therefore consistent with a Ferromagnetic instability (infinite uniform spin susceptibility down to $T=0$) as far as I can judge.

Clearly not a gapped spin liquid as the spin susceptibility should vanish exponentially in T for $T \rightarrow 0$, but not even a gapless projected wavefunction could be compatible with infinite or very large uniform spin susceptibility.

The other effects presented by the authors seem to be rather irrelevant as compared to their main result if these data do not have to be corrected by some background or whatever technicality, (e.g. strongly temperature dependent

form factor). The authors should therefore spend some effort to rule out the

rather unexpected ferromagnetic instability, that is the main evidence I find from their data.

In any case the results are surprising, either there is a ferromagnetic instability or we should have a gapped short range RVB, according to the authors.

In my opinion it is therefore possible that this material is not well described by the Heisenberg model (expected to be antiferromagnetically ordered in its ground state) but by an Hubbard model at moderate U/t value.

Can the author estimate the U/t value?

In conclusion I do not recommend publication of this manuscript in the present form.

Reviewer #2 (Remarks to the Author):

The present manuscript is a state-of-the-art experimental study of YbMgGaO₄ by means of inelastic neutron scattering on single-crystal samples. This material has attracted (and continues to attract) enormous interest in the magnetism community as a possible quantum spin-liquid candidate on the triangular-lattice. As a matter of fact, two of the present authors are to be credited with the identification of possible exotic physics in this material, as reported in Scientific Reports and PRL in 2015/2016. One of the appeal of YbMgGaO₄ is that it potentially marries geometrical frustration and spin-orbit coupling to stabilize a quantum entangled state of matter with exotic excitations. The exact origin of this behavior remains puzzling and this manuscript is a valuable contribution to the scientific debate on the mechanism(s) behind YbMgGaO₄ behavior.

In December 2016, two separate groups published related inelastic neutron scattering results on YbMgGaO₄ in Nature (Shen et al., Ref 16) and Nature Physics (Paddison et al., Ref 17). Two of the co-authors of the present work are also co-authors on the Shen et al. work, but presumably for providing help with sample characterization. The present study can thus be considered a third and independent study of this topical material with neutrons. In my opinion, the existence of the above two works does not disqualify this work for publication in Nature Communications, provided it contains novel experimental results or interpretation. The main experimental value of this work compared to the above studies is the extension of the dynamic range of the experiment to energies below 0.2 meV and reaching as low as 0.02 meV. This is an interesting regime to track for the existence of a gap and the authors argue for the presence of two excitations regime: above and below 0.13 meV. From their analysis, the authors argue that the excitations above 0.13 meV resemble a RVB state (Resonating Valence Bond -- proposed by Anderson in the 70's for the triangular Heisenberg model) while not much is said of the spatial correlations of the signal below that energy.

Overall, I feel the data is of good quality and the paper relatively easy to read. However, I also find some important flaws that prevent me from recommending publication in Nature Communications:

1) The authors resort to the "equal-time" structure factor of nearest-neighbor valence-bonds to model their data. However, they compare this model to a high-energy "slice" through their data ($0.65 < E < 0.75$ meV) or a "cut" integrated only from 0.5 meV. The comparison with the "equal-time" theory is thus not valid: the data should be integrated upon the entire dynamic range ($0 < E < 1.5-2$ meV). The claim made in the introduction that RVB correlations account for the spectral weight above 0.1 meV is therefore not substantiated. It would be beneficial to show how the spatial correlations look like below 0.5 meV, and also below 0.1 meV

2) The authors associate the claim of a "spinon Fermi surface" in YbMgGaO₄ to Shen et al. (Ref. 16) and Paddison et al. (Ref. 17). This scenario is certainly not discussed nor favored in Ref. 17, where the role of second-neighbor interactions and disorder is brought up. Generally speaking, the authors should place their work in the context of the above two papers. In particular, do the authors uncover elastic scattering anywhere in the triangular Brillouin zone? How does the momentum dependence of the signal below 0.13 meV look like?

3) The persistence of excitations to high temperatures and the opening of a gap by magnetic field are already discussed in Ref. 17. I feel the authors should focus more on the original discovery of this paper, namely the presence of two energy scales.

4) Other triangular lattice materials with RVB correlations have been proposed and explored experimentally, see for instance $\text{LiZn}_2\text{Mo}_3\text{O}_8$ (doi:10.1038/nmat3329).

Reviewer #3 (Remarks to the Author):

I have read through this manuscript which reports on a sophisticated inelastic neutron scattering experiment on YbMgGaO_4 , which is a topical candidate for a quantum spin liquid (QSL) state. Although the quality of the data is very high, I feel that the manuscript has several very significant deficiencies and should not be published in its present form. I elaborate below.

1. The authors make the claim that the experimental data supports evidence for an RVB like state with singlets formed (presumably between $S_{\text{eff}}=1/2$ moments on the Yb sites) at the near-neighbour level on the triangular lattice and beyond. The authors claim is a very strong one: on page 2 of the manuscript states that the high energy excitations can be "unambiguously ascribed to nearest neighbour RVB". However there is very little quantitative analysis of the high quality data to support such a claim. The authors fit some of the Q-dependence of the scattering, which is concentrated along Brillouin zone boundaries, to Eq. 1, but which contains geometric terms only. Apart from a magnetic form factor, I don't believe this form uniquely identifies the scattering as due to singlet-triplet transitions within an RVB model - it probably only refers to near neighbour correlations of some sort. Other arguments are made, but they are more qualitative and I feel the conclusions drawn are too strong for the quality of the interpretation of the data.

2. There is no information given as to how the $S=1/2$ moments arise from the Yb^{3+} sites, and how these participate in RVB type singlets. These are not the spin only moments of Cu^{2+} that are found in La_2CuO_4 and Herbertsmithite - they originate for total angular momentum J , which is then split by crystal fields. Also, the nature of the exchange interactions in such a system, with high spin-orbit coupling, is very likely not simple Heisenberg exchange - rather it is very likely anisotropic exchange, as studied for example by Ross et al in $\text{Yb}_2\text{Ti}_2\text{O}_7$ (PRX Phys. Rev. X 1, 021002, 2011). How do these $S=1/2$ entities form traditional singlets? This needs to be discussed as the "simple" case that applied to, say Herbertsmithite, does not apply here.

3. Related to point 2, what is the crystal field ground state of Yb^{3+} in this material? Has this been studied? This determines both the moment size and anisotropy, so it is pretty important as a starting point in the description of the system.

4. The authors make no mention of possible disorder effects in this material, yet it is very likely that there is very appreciable disorder in this system as the Ga^{3+} and Mg^{2+} do not differ much in either charge or ionic size - so they almost certainly mix at some level. It is even possible that they are fully disordered. The presence or absence of disorder is a key point in the discussion of other singlet ground state systems such as Herbertsmithite. It could also profoundly effect the crystal field states of Yb^{3+} as occurs with weak disorder in $\text{Yb}_2\text{Ti}_2\text{O}_7$ (see Gaudet et al, PRB 92, 134420, 2015).

5. I found some of the phrasing in the manuscript confusing. For example in the first paragraph, following the abstract, the authors write: "Recently a triangular QSL candidate YbMgGaO_4 attracted much interest [3-5], because it is free from magnetic defects[10-11], spatial coupling anisotropy[11-13] and antisymmetric D-M anisotropy [14]." When I read this, I thought the authors were saying that there is experimental evidence to show that YbMgGaO_4 is free from all these effects. But the references [10 -14] refer to other materials, not to YbMgGaO_4 . While the authors were not trying to do so, I found this sentence to be misleading.

Reviewer #1 (Remarks to the Author):

General Comment: In this work the authors present a new interpretation of the data of INS based on more accurate and better resolved experiments in YbMgGaO_4 , a material that according to the authors should represent a rather clean realization of the isotropic Heisenberg model on the triangular lattice.

The results may be interesting, but in my opinion the interpretation of the data is not as clear as the authors claim. The authors state that Eq.1 univocally identifies the INS spectrum of a short range RVB. However I have several doubts. Any smooth function of the momenta, satisfying the spatial symmetry of the triangular lattice and that vanishes at the Gamma point (as a consequence of spin conservation at $T=0$) can be approximated by an expression like that (just because higher harmonics are much weaker in a smooth function expected at high T). At high temperature this is certainly the fingerprint that in this material the spins lie in a triangular lattice (apart of the fact that this function does no longer need to vanish at Gamma), but nothing more than that. The surprising fact is that the authors find that this ansatz for the INS is essentially valid also at very low temperatures (0.1K about 150 times smaller than the expected J_0). This is certainly interesting but I found the data hard to believe and to understand (e.g. the momenta notations) at low temperatures.

Reply: We thank the Reviewer for his/her interest on our work, and for fruitful comments that helped us to improve the quality of our work. Before addressing the specific points raised by the Reviewer, we would like to clarify why our results show more than merely "spins that lie on a triangular lattice". Below, re-iterate our arguments that support the RVB scenario:

(1) The \mathbf{Q} -dependence of the signal follows the nearest-neighbor RVB model on the triangular lattice very well, and it is essentially inconsistent with other disordered two-dimensional (2D) models, where long-range antiferromagnetic correlations dominate (for example, the next- and third-nearest-neighbor RVB models on the triangular lattice), see Figure R4.

(2) Temperature dependence of the excitations follows the RVB scenario, see our reply to Reviewer #3 below.

(3) No spin freezing was observed, according to our previously reported muon spin relaxation work [Phys. Rev. Lett. 117, 097201 (2016)]. Therefore, any static state with nearest-neighbor correlations can be excluded, and a dynamic state with purely nearest-neighbor correlations, i.e., an RVB state is observed.

(4) The isotropic coupling is antiferromagnetic, according to our previous characterization, [Phys. Rev. Lett. 115, 167203 (2015)], and antiferromagnetic couplings are a crucial ingredient of Anderson's RVB scenario.

Comment-1: First let me discuss about the notations. In Eq.1 the signal vanishes correctly for $H=K=0$ but not for $H,K=-0.5$ as is instead reported in Fig.1f. It looks to me that Eq.1 is not typed well and should be substituted by the one reported in the supplementary information (Eq.S2).

Reply: In Fig. 1f, as well as in Fig. 1a and b, the coordinate point (x, y) represents the wave-vector, $x[1, 1, 0] + y[1, -1, 0] = [x+y, x-y, 0] = (x+y)\beta_1+(x-y)\beta_2+0\beta_3$ (here, β_1 , β_2 and β_3 are the inverted lattice vectors). That is, $H = x+y$ and $K = x-y$ in Eq. (2) (see the updated version).

The point $(-0.5, -0.5)$ (Γ_2) in Fig. 1f corresponds to $H = -1$ and $K = 0$ in Eq. (2), so the intensity should vanish just as Fig. 1f shows. Conversely, $H = -0.5$ and $K = -0.5$ in Eq. (2) corresponds to

the point (-0.5, 0) in Fig. 1f, and the intensity doesn't vanish. There is no contradiction between Eq. (2) and Fig. 1f.

The similar representation is used in other recent publications, such as Nature 540, 559–562 (2016) (triangular lattice) and Nature 492, 406-410 (2012) (kagome lattice). The corresponding equal-time INS intensity of uncorrelated nearest-neighbor valence bonds on a *kagome* lattice is proportional to

$$|F(\mathbf{Q})|^2 = \frac{2}{3} |f(\mathbf{Q})|^2 \{3 - \cos(\pi H) - \cos(\pi K) - \cos[\pi(H+K)]\}. \quad \text{Eq. R1}$$

For herbertsmithite, $f(\mathbf{Q})$ stands for the magnetic form factor of free Cu^{2+} , Eq. R1 is exactly consistent with the calculations previously reported in Fig. 1e, Fig. 3b (solid line), and Fig. 3c (solid line) of the reference, Nature 492, 406-410 (2012). We had checked out the above analyses carefully.

The above discussions (explanations) were included in the revised version.

Comment-2: *Second, assuming the above is correct the experimental data are also hard to believe. The panels a and b are qualitatively different apart for colors from the panel f. At Gamma there is a huge signal both in a and b. The fact that they are colored in the same way of panel f is a bit confusing.*

Reply: Experimental INS intensity has a minimum at the Gamma points, and this value should be treated simply as background. In the case of YbMgGaO_4 , the non-zero INS intensity at the Gamma points has also been reported in Fig. 4a and 4b of Nature 540, 559–562 (2016) and in Fig. 2d of Nat. Phys. 13, 117 (2017). Our RVB calculation does not show this background, because this calculation refers to zero temperature, whereas experimental data are collected at a non-zero temperature. Its relation to other observables is further explained in our reply to *Comment-3*. This point is discussed in the revised manuscript.

Comment-3: *As in the ansatz, at $T=0$ the signal should be 0 at Gamma if we are dealing with a spin liquid, namely at $(K,H) = (0,0)$ or $(-0.5,-0.5)$ in their way to represent the BZ. At finite temperature, if the spin susceptibility is assumed to be at most finite at $T=0$ the signal at Gamma should go down to zero at least as T , whereas the signal at Gamma increases at $T=0.1\text{K}$ (!) as compared to the $T=35\text{K}$. The data look therefore consistent with a Ferromagnetic instability (infinite uniform spin susceptibility down to $T=0$) as far as I can judge. Clearly not a gapped spin liquid as the spin susceptibility should vanish exponentially in T for $T \rightarrow 0$, but not even a gapless projected wavefunction could be compatible with infinite or very large uniform spin susceptibility. The other effects presented by the authors seem to be rather irrelevant as compared to their main result if these data do not have to be corrected by some background or whatever technicality, (e.g. strongly temperature dependent form factor). The authors should therefore spend some effort to rule out the rather unexpected ferromagnetic instability, that is the main evidence I find from their data.*

In any case the results are surprising, either there is a ferromagnetic instability or we should have a gapped short range RVB, according to the authors. In my opinion it is therefore possible that this material is not well described by the Heisenberg model (expected to be antiferromagnetically ordered in its ground state) but by an Hubbard model at moderate U/t value. Can the author estimate the U/t value?

In conclusion I do not recommend publication of this manuscript in the present form.

Reply: We thank the Reviewer for this important comment.

At the Gamma point (Γ_1 ; $K = 0$; $H = 0$, and $Q = 0$), the measured INS intensity, $I_\Gamma(E)$, is related to the uniform spin susceptibility, $\chi'(E)$, per the Reviewer's comment. Via the fluctuation-dissipation theorem, we obtain imaginary part of the susceptibility [Rev. Sci. Instrum. 84, 083906 (2013)],

$$\chi''(E) \propto \left(1 - e^{-\frac{E}{k_B T}}\right) I_\Gamma(E), \quad \text{Eq. R2}$$

assuming the Debye-Waller factor is temperature-independent below ~ 40 K.

Real part of the susceptibility can be obtained through the Kramers-Kronig transformation,

$$\chi'(E) \propto \frac{1}{\pi} \int \frac{\left(1 - e^{-\frac{E'}{k_B T}}\right) I_\Gamma(E')}{E' - E} dE'. \quad \text{Eq. R3}$$

Figure R1. Energy dependences of the uniform susceptibilities ($Q \sim 0$), as measured at 0.1 and 35 K. **a**, imaginary part of the susceptibility; **b**, real part of the susceptibility. The elastic signal determined by fitting the spectrum at $|E| < \sigma$ was subtracted from $I_\Gamma(E)$. The spectra at $0.02 < E < 0.6$ meV were obtained from the measurements with $E_i = 1.26$ meV ($\sigma = 0.02$ meV), while the spectra at $E > 0.6$ meV were obtained from the measurements with $E_i = 5.5$ meV ($\sigma = 0.16$ meV). The integration in the Kramers-Kronig relation was performed starting from 0.02 meV, and the data points with $|E' - E| < 0.005$ meV were excluded from the integral to avoid the divergence (see Eq. R3).

Figure R2. Temperature dependences of the static bulk susceptibilities measured parallel ($\chi_{||}$) and perpendicular (χ_{\perp}) to the c -axis for the YbMgGaO_4 single crystal [Phys. Rev. Lett. 117, 097201 (2016)]. The black stars show the uniform spin susceptibilities at $E = 0.04$ meV, while the red stars show the uniform spin susceptibilities around $E = 0.7$ meV.

Energy dependence of the uniform ($\mathbf{Q} \sim 0$) susceptibility is shown in Fig. R1. Experimental INS intensity (I_r) at 0.1 K is significantly larger than that at 35 K, $I_r(0.1 \text{ K}) \sim 2I_r(35 \text{ K})$ ($0.5 \leq E \leq 1.5$ meV), per the Reviewer's comment. Nevertheless, above 0.5 meV real part of the susceptibility (χ') measured at 0.1 K turns out to be zero within the error bar, which is well in line with our RVB scenario. High values of χ' are seen at energies below 0.15 meV only, indicating some low-energy physics beyond the simple RVB picture.

Although the high static bulk susceptibility at 0.48 K may indicate a ferromagnetic instability, as the Reviewer suggested, we find this scenario very unlikely, given purely antiferromagnetic couplings in YbMgGaO_4 . Our magnetization measurements down to 0.5 K do not show any signatures of remnant magnetization [Scientific Reports 5, 16419 (2015)], whereas μSR excludes any long-range order that a ferromagnetic instability would induce [Phys. Rev. Lett. 117, 097201 (2016)].

Further insight can be obtained from comparing I_r with bulk magnetic susceptibility χ . While experimental χ increases within the temperature range of our measurement, down to 0.48 K, the susceptibility value expected from the INS intensity at 0.1 K is much lower than the value at 0.48 K. Therefore, there has to be a downward trend in the susceptibility between 0.48 K and 0.1 K, which excludes any ferromagnetic instability in our system.

We revised the manuscript following the Reviewer's comment: i) we explain that uniform susceptibility extracted from I_r is consistent with the RVB scenario above 0.5 meV; ii) using experimental temperature-dependent susceptibility we argue that the relatively high value of I_r at 0.1 K is not a signature of a ferromagnetic instability.

Finally, we would like to comment on the U/t ratio. For YbMgGaO_4 , the excellent transparency (band gap > 3 eV) and insulating behavior (the resistance is unmeasurably large, $> 20 \text{ M}\Omega$) of the single crystal suggest the charge gap is quite large, thus U is very large too. Since the $4f$ electrons of Yb^{3+} are extremely well localized, t should be extremely small. As a result, U/t should be very large, placing YbMgGaO_4 deep in the Mott insulator regime of the Hubbard model.

Reviewer #2 (Remarks to the Author):

General Comment: *The present manuscript is a state-of-the-art experimental study of YbMgGaO₄ by means of inelastic neutron scattering on single-crystal samples. This material has attracted (and continues to attract) enormous interest in the magnetism community as a possible quantum spin-liquid candidate on the triangular-lattice. As a matter of fact, two of the present authors are to be credited with the identification of possible exotic physics in this material, as reported in Scientific Reports and PRL in 2015/2016. One of the appeal of YbMgGaO₄ is that it potentially marries geometrical frustration and spin-orbit coupling to stabilize a quantum entangled state of matter with exotic excitations. The exact origin of this behavior remains puzzling and this manuscript is a valuable contribution to the scientific debate on the mechanism(s) behind YbMgGaO₄ behavior.*

In December 2016, two separate groups published related inelastic neutron scattering results on YbMgGaO₄ in Nature (Shen et al., Ref 16) and Nature Physics (Paddison et al., Ref 17). Two of the co-authors of the present work are also co-authors on the Shen et al. work, but presumably for providing help with sample characterization. The present study can thus be considered a third and independent study of this topical material with neutrons. In my opinion, the existence of the above two works does not disqualify this work for publication in Nature Communications, provided it contains novel experimental results or interpretation. The main experimental value of this work compared to the above studies is the extension of the dynamic range of the experiment to energies below 0.2 meV and reaching as low as 0.02 meV. This is in interesting regime to track for the existence of a gap and the authors argue for the presence of two excitations regime: above and below 0.13 meV. From their analysis, the authors argue that the excitations above 0.13 meV resemble a RVB state (Resonating Valence Bond -- proposed by Anderson in the 70's for the triangular Heisenberg model) while not much is said of the spatial correlations of the signal below that energy.

Overall, I feel the data is of good quality and the paper relatively easy to read. However, I also find some important flaws that prevent me from recommending publication in Nature Communications:

Reply: We thank the Reviewer for the very positive appraisal on our work and very helpful comments.

Comment-1: *The authors resort to the "equal-time" structure factor of nearest-neighbor valence-bonds to model their data. However, they compare this model to a high-energy "slice" through their data ($0.65 < E < 0.75 \text{ meV}$) or a "cut" integrated only from 0.5 meV. The comparison with the "equal-time" theory is thus not valid: the data should be integrated upon the entire dynamic range ($0 < E < 1.5\text{-}2 \text{ meV}$). The claim made in the introduction that RVB correlations account for the spectral weight above 0.1 meV is therefore not substantiated. It would be beneficial to show how the spatial correlations look like below 0.5 meV, and also below 0.1 meV.*

Reply: We thank the Reviewer for this caveat. However, we have to admit that getting the data in the whole energy range is not an easy task. A sufficiently broad \mathbf{Q} range can be accessed with the incident energy of $E_i = 5.5 \text{ meV}$ only. In Fig. R3, we show energy dependence of the integrated neutron intensity from this measurement. The data below 0.5 meV are contaminated by the elastic signal. Above 0.5 meV, one clearly sees a very broad hump that extends to $\sim 2 \text{ meV}$. The "peak" of this hump is around $\sim 0.7 \text{ meV}$, and therefore we chose the $0.65 \leq E \leq 0.75$

meV slice for the analysis. However, essentially the same result is obtained for the data integrated from 0.5 to 1.5 meV, see Fig. R4. Thus the RVB correlations indeed account for the spectral weight in this energy range.

For herbertsmithite, the INS data integrated from 1 meV (also *non-zero*) to 11 meV was used to compare with the similar calculated *equal-time* structure factor (see Eq. R1) [Nature 492, 406-410 (2012)].

Figure R3. Energy dependence of the integrated neutron scattering intensity for YbMgGaO_4 with $E_i = 5.5$ meV. Solid line represents the elastic contribution, extracted from the data at $|E| < \sigma$ by a fit to a convoluted Lorentzian-Gaussian peak function.

Figure R4. Wave-vector dependence of the INS intensity along $[-1, K+0.5, 0]$, $[H, 0, 0]$, and $[-H, 0.5H, 0]$, with lines representing different RVB calculations, $a|F(\mathbf{Q})|^2+b$. Here, a and b are fitted constants for the proportionality and background, respectively. Please note: Due to an unequal number of integrated data points, both a and b are slightly different along different \mathbf{Q} directions. Marginal boundary effects can be seen at the beginning and end of each series of the experimental data.

Lower energies have to be probed with $E_i = 1.26$ meV (see Fig. R6) that, however, does not give access to the sufficiently broad \mathbf{Q} -range, and even the first Brillouin zone boundary is hardly

reached (see Fig. R7). There are no qualitative differences between the spectral cuts at ~ 0.3 and ~ 0.7 meV in all measured \mathbf{Q} space ($E_i = 1.26$ meV) apart from an overall increase in the intensity (please compare b with c at 35 K and e with f at 0.1 K, in Fig. R7). Our statement that the RVB correlations account for the spectral weight down to 0.1 meV was perhaps somewhat exaggerated without considering the energy-dependent spectra, but the interpretation of the spectral weight above 0.5 meV is robust, as shown in Fig. R4, and the manuscript has been revised accordingly. The complete \mathbf{Q} -dependent analysis for energies below 0.5 meV would require a measurement with a better coverage of the \mathbf{Q} -space.

Given the Reviewer's request, in Figs. R5-R7 we also show the data at lower energies. Unfortunately, they are contaminated by a polycrystalline-like signal of some sort (see Figs. R5, R7 a and d). It may originate from the magnet, cryostat, and/or sample holder. We were unable to get rid of this signal within the beamtime available for this experiment.

Figure R5. Wave-vector dependence of the INS intensity ($0.25 \leq E \leq 0.35$ meV, with $E_i = 5.5$ meV) measured **a**, at 35 K and 0 T, **b**, at 0.1 K and 0 T.

Figure R6. Energy dependence of the integrated neutron scattering intensity for YbMgGaO_4 with $E_i = 1.26$ meV. Solid line represents the elastic contribution. The blue stripes denote different energy cuts shown in Fig. R7.

Figure R7. Wave-vector dependence of the INS intensity measured under 0 T with $E_i = 1.26$ meV **a**, at $0.07 \leq E \leq 0.11$ meV and 35 K, **b**, at $0.28 \leq E \leq 0.32$ meV and 35 K, **c**, at $0.68 \leq E \leq 0.72$ meV and 35 K, **d**, at $0.07 \leq E \leq 0.11$ meV and 0.1 K, **e**, at $0.28 \leq E \leq 0.32$ meV and 0.1 K, and **f**, at $0.68 \leq E \leq 0.72$ meV and 0.1 K.

Comment-2: The authors associate the claim of a "spinon Fermi surface" in YbMgGaO₄ to Shen et al. (Ref. 16) and Paddison et al. (Ref. 17). This scenario is certainly not discussed nor favored in Ref. 17, where the role of second-neighbor interactions and disorder is brought up. Generally speaking, the authors should place their work in the context of the above two papers. In particular, do the authors uncover elastic scattering anywhere in the triangular Brillouin zone? How does the momentum dependence of the signal below 0.13 meV look like?

Reply: Okay, we removed the reference (Paddison et al.) there. We present the elastic scattering signals ($-0.1 \leq E \leq 0.1$ meV) in the Brillouin zone (see Fig. R8). The momentum dependences of the signal below 0.13 meV have been shown in Fig. R7 a and d. As we have discussed, the polycrystalline signal of some sort seriously contaminates the intrinsic signal of YbMgGaO₄. We tried to subtract the 35 K signal from the 0.1 K signal, but this subtraction produces negative intensities below 0.13 meV in some regions of the \mathbf{Q} space (see Fig. R8c, f and Fig. R9), the result is quite noisy, and the \mathbf{Q} space measured with $E_i = 1.26$ meV is small (see Fig. R8f and Fig. R9). Therefore, we refrain from any analysis of these data.

The above discussions were included in the revised Supplementary Information.

Figure R8. Wave-vector dependence of the elastic scattering signal (integrated over $-0.1 \leq E \leq 0.1$ meV) **a.** with $E_i = 5.5$ meV at 0.1 K, **b.** with $E_i = 5.5$ meV at 35 K, **d.** with $E_i = 1.26$ meV at 0.1 K, and **e.** with $E_i = 1.26$ meV at 35 K. **I(0.1 K) - I(35 K)** **c.** with $E_i = 5.5$ meV, and **f.** with $E_i = 1.26$ meV.

Figure R9. Wave-vector dependences of the subtracted INS intensity $[I(0.1K) - I(35K)]$ integrated over $0.07 \leq E \leq 0.11$ meV with $E_i = 1.26$ meV.

Comment-3: The persistence of excitations to high temperatures and the opening of a gap by magnetic field are already discussed in Ref. 17. I feel the authors should focus more on the original discovery of this paper, namely the presence of two energy scales.

Reply: Okay, we have revised our statements in the updated version.

Comment-4: Other triangular lattice materials with RVB correlations have been proposed and explored experimentally, see for instance $\text{LiZn}_2\text{Mo}_3\text{O}_8$ (doi:10.1038/nmat3329).

Reply: We thank the Reviewer for providing the important experimental example, $\text{LiZn}_2\text{Mo}_3\text{O}_8$. It has been discussed in the revised manuscript, and the related references were also cited.

Reviewer #3 (Remarks to the Author):

General Comment: I have read through this manuscript which reports on a sophisticated inelastic neutron scattering experiment on YbMgGaO_4 , which is a topical candidate for a quantum spin liquid (QSL) state. Although the quality of the data is very high, I feel that the manuscript has several very significant deficiencies and should not be published in its present form. I elaborate below.

Reply: We thank the Reviewer for his/her interest on our work, and fruitful comments.

Comment-1: The authors make the claim that the experimental data supports evidence for an RVB like state with singlets formed (presumably between $S_{\text{eff}}=1/2$ moments on the Yb sites) at the near-neighbour level on the triangular lattice and beyond. The authors claim is a very strong one: on page 2 of the manuscript states that the high energy excitations can be "unambiguously ascribed to nearest neighbour RVB". However there is very little quantitative analysis of the high quality data to support such a claim. The authors fit some of the Q-dependence of the scattering, which is concentrated along Brillouin zone boundaries, to Eq. 1, but which contains geometric terms only. Apart from a magnetic form factor, I don't believe this form uniquely identifies the scattering as due to singlet-triplet transitions within an RVB model - it probably only refers to near neighbour correlations of some sort. Other arguments are made, but they are more qualitative and I feel the conclusions drawn are too strong for the quality of the interpretation of the data.

Reply: Our analysis of the \mathbf{Q} -dependence of the integrated neutron intensity indeed contains only geometrical terms and the magnetic form-factor. Nevertheless, this analysis (see Fig. R4) essentially excludes any correlations beyond nearest neighbors, which is a quite unusual result on its own. To further verify that the excitations are related to singlet-"triplet" (in this case, the "triplet" is different from a simple triplet for Heisenberg spins, see below) transitions, we analyzed their temperature dependence.

The fitted constants for the proportionality, $a(35\text{K})/a(0.1\text{K}) \sim 0.3$ (see Table R1), are consistent with the expected ratio,

$$\frac{N(T_2 = 35\text{K})}{N(T_1 = 0.1\text{K})} \sim \frac{1 + \exp\left(-\frac{0.809J_0}{k_B T_1}\right) + \exp\left(-\frac{1.012J_0}{k_B T_1}\right) + \exp\left(-\frac{1.179J_0}{k_B T_1}\right)}{1 + \exp\left(-\frac{0.809J_0}{k_B T_2}\right) + \exp\left(-\frac{1.012J_0}{k_B T_2}\right) + \exp\left(-\frac{1.179J_0}{k_B T_2}\right)} \sim 0.26, \quad \text{Eq. R4}$$

where individual energies correspond to eigenstates of a spin dimer with anisotropic interactions (see our response to the next comment), and J_0 is isotropic exchange coupling. With increasing temperature, a larger fraction of nearest-neighbor singlets is excited.

Table R1. The fitted constants for the proportionality and background at 35 and 0.1 K using the nearest-neighbor RVB model.

Directions	[-1, K+0.5, 0] (Fig.1c)		[H, 0, 0] (Fig.1d)		[-H, 0.5H, 0] (Fig.1e)	
Temperature	35 K	0.1 K	35 K	0.1 K	35 K	0.1 K
a	29.1(1.7)	109.6(2.7)	50.9(2.2)	158.3(4.6)	42.5(2.1)	126.4(3.9)
a(35K)/a(0.1K)	0.266(22)		0.322(23)		0.336(27)	
b or I_{Γ}	165.4(3.2)	328.6(4.9)	130.7(4.3)	286.1(8.5)	118.6(5.1)	272.5(9.3)
b(35K)/b(0.1K) or $I_{\Gamma}(35\text{K})/I_{\Gamma}(0.1\text{K})$	0.503(17)		0.457(29)		0.435(34)	

Additionally, our previous muon spin relaxation data effectively rule out any kind of frozen state with nearest-neighbor correlations. Altogether, our claim for the RVB nature of excitations above 0.5 meV is supported by four main arguments:

(1) The \mathbf{Q} -dependence of the signal follows the nearest-neighbor RVB model on the triangular lattice very well, and it is essentially inconsistent with other disordered two-dimensional (2D) models, where long-range antiferromagnetic correlations dominate (for example, the next- and third-nearest-neighbor RVB models on the triangular lattice), see Figure R4.

(2) The fitted constants for the proportionality, $a(35\text{K})/a(0.1\text{K}) \sim 0.3$ (please see Table R1), are consistent with the expected ratio ~ 0.26 (see Eq. R4).

(3) No spin freezing was observed, according to our previously reported muon spin relaxation work [Phys. Rev. Lett. 117, 097201 (2016)].

(4) The isotropic coupling is antiferromagnetic, according to our previous characterization. [Phys. Rev. Lett. 115, 167203 (2015)].

Following the Reviewer's comment, we removed the word "unambiguously", because it seems impossible to exclude all possible non-RVB states. However, we believe that we have strong evidence that the nearest-neighbor RVB scenario holds for this material.

Comment-2: *There is no information given as to how the $S=1/2$ moments arise from the Yb^{3+} sites, and how these participate in RVB type singlets. These are not the spin only moments of Cu^{2+} that are found in La_2CuO_4 and Herbertsmithite - they originate for total angular momentum J , which is then split by crystal fields. Also, the nature of the exchange interactions in such a system, with high spin-orbit coupling, is very likely not simple Heisenberg exchange - rather it is very likely anisotropic exchange, as studied for example by Ross et al in $\text{Yb}_2\text{Ti}_2\text{O}_7$ (PRX Phys. Rev. X 1, 021002, 2011). How do these $S=1/2$ entities form traditional singlets? This needs to be discussed as the "simple" case that applied to, say Herbertsmithite, does not apply here.*

Reply: We agree with the Reviewer, and discuss this issue in the revised manuscript. The reference [Phys. Rev. X 1, 021002 (2011)] has also been cited. Yes, the effective spin-1/2 moment of Yb^{3+} is different from that of Cu^{2+} . Owing the strong spin-orbit coupling, magnetic couplings in YbMgGaO_4 are highly anisotropic in the spin space. This aspect has been addressed in our previous work [Phys. Rev. Lett. 115, 167203 (2015)]. Intuitively, the anisotropy of the couplings may alter the singlet state, as the Reviewer kindly points out. The ground state of the Yb^{3+} dimer may not be a singlet, and the excited states are no longer three degenerate triplets.

The effective spin-1/2 Hamiltonian between two nearest-neighbor Yb^{3+} spins (\mathbf{S}_1 and \mathbf{S}_2) had been reported in our previous publication [Phys. Rev. Lett. 115, 167203 (2015)].

$$\mathbf{H} = J_{zz}S_1^zS_2^z + J_{\pm}(S_1^+S_2^- + S_1^-S_2^+) + J_{\pm\pm}(\gamma_{12}S_1^+S_2^+ + \gamma_{12}^*S_1^-S_2^-) - \frac{iJ_{z\pm}}{2}(\gamma_{12}^*S_1^+S_2^z - \gamma_{12}S_1^-S_2^z + \langle 1 \leftrightarrow 2 \rangle). \quad \text{Eq. R5}$$

Here, $J_{\pm} \sim 0.9$ K, $J_{zz} \sim 0.98$ K, $J_{z\pm} \sim 0$ K, $J_{\pm\pm} \sim (+ \text{ or } -)0.155$ K, and $\gamma_{12} = 1, e^{i2\pi/3}, e^{-i2\pi/3}$ for the bonds along the $\mathbf{a}_1, \mathbf{a}_2, \mathbf{a}_3$ directions (see Fig. R10), respectively.

Figure R10. Triangular lattice of Yb^{3+} in YbMgGaO_4 .

Through the diagonalization of the Hamiltonian (Eq. R5), we find that the ground state is the **strict** traditional singlet, $\frac{1}{\sqrt{2}}(|\uparrow\downarrow\rangle - |\downarrow\uparrow\rangle)$, and is independent on the bond direction (\mathbf{a}_1 , \mathbf{a}_2 , or \mathbf{a}_3) and the anisotropic coupling parameters, as long as the isotropic coupling, $J_0 \equiv (4J_{\pm} + J_{zz})/3$, is antiferromagnetic (i.e. $J_0 > 0$). And $J_0 = 1.5(1)$ K or $0.13(1)$ meV > 0 had been safely determined in our previous studies [Phys. Rev. Lett. 115, 167203 (2015)].

The energy gaps from the ground-state singlet (with the energy eigenvalue of $-3/4J_0$) to the three excited states are also independent of the bond direction (\mathbf{a}_1 , \mathbf{a}_2 , or \mathbf{a}_3), but are strongly dependent on the anisotropic coupling parameters. Since $J_{z\pm}$ had been measured to be almost zero [Phys. Rev. Lett. 115, 167203 (2015)], all of the three excited energies (from $-3/4J_0$) are independent on the sign of $J_{\pm\pm}$ at $J_{z\pm} = 0$, and are calculated to be $0.809J_0$, $1.012J_0$, and $1.179J_0$, respectively.

As a result, the uncorrelated spin-1/2 nearest-neighbor valence bond model, which had been used for herbertsmithite [Nature 492, 406-410 (2012)], is applicable for YbMgGaO_4 as well.

We regret that this conceptual analysis was missing in the previous version of our manuscript and thank the Reviewer for mentioning this point.

Comment-3: *Related to point 2, what is the crystal field ground state of Yb^{3+} in this material? Has this been studied? This determines both the moment size and anisotropy, so it is pretty important as a starting point in the description of the system.*

Reply: The crystal field ground state of Yb^{3+} in YbMgGaO_4 has been studied in our recent high-energy INS work [arxiv: 1702.01981 (accepted in Phys. Rev. Lett.)]. The wavefunctions of the ground-state Kramers doublet are $\sim \pm 0.71|\pm 7/2\rangle \mp 0.36|\mp 5/2\rangle + 0.60|\pm 1/2\rangle$. The effective spin-1/2 g-factors (g_{\parallel} and g_{\perp}) had also been reported in our previously reported work [Phys. Rev. Lett. 115, 167203 (2015)]. The information on the moment size and anisotropy, and the wavefunctions of the ground-state Kramers doublet was included in the revised Supplementary Information.

Comment-4: *The authors make no mention of possible disorder effects in this material, yet it is very likely that there is very appreciable disorder in this system as the Ga^{3+} and Mg^{2+} do not differ much in either charge or ionic size - so they almost certainly mix at some level. It is even possible that they are fully disordered. The presence or absence of disorder is a key point in the discussion of other singlet ground state systems such as Herbertsmithite. It could also profoundly effect the crystal field states of Yb^{3+} as occurs with weak disorder in $\text{Yb}_2\text{Ti}_2\text{O}_7$ (see Gaudet et al, PRB 92, 134420, 2015).*

Reply: The disorder may be important indeed. Our CEF study provides first indications for the relevance of the Mg/Ga disorder as well as the microscopic analysis of its origin [arxiv: 1702.01981 (accepted in Phys. Rev. Lett.)]. The low-energy INS study presented in this manuscript is more phenomenological. Our main goal is to establish the unexpected RVB physics in YbMgGaO₄. The disorder may be one of its ingredients, but quantifying effects of disorder is by no means an easy task, as demonstrated by the existing models of herbertsmithite, which is far from being completely understood despite at least 10 years of intensive research. We mention the disorder in the revised version of the manuscript, but we believe that it will be premature to analyze its effect on magnetic excitations before we know how individual exchange couplings are affected by the disorder.

We thank the Reviewer for providing the important reference [Gaudet et al, Phys. Rev. B 92, 134420 (2015)]. It has been discussed and cited in the revised manuscript.

Comment-5: *I found some of the phrasing in the manuscript confusing. For example in the first paragraph, following the abstract, the authors write: "Recently a triangular QSL candidate YbMgGaO₄ attracted much interest [3-5], because it is free from magnetic defects[10-11], spatial coupling anisotropy[11-13] and antisymmetric D-M anisotropy [14]." When I read this, I thought the authors were saying that there is experimental evidence to show that YbMgGaO₄ is free from all these effects. But the references [10 -14] refer to other materials, not to YbMgGaO₄. While the authors were not trying to do so, I found this sentence to be misleading.*

Reply: Okay, this sentence has been modified according to the Reviewer's suggestion.

Reviewers' comments:

Reviewer #1 (Remarks to the Author):

The authors have replied in a satisfactory manner most of my concerns.

Before publication I recommend only to elaborate on the meaning of an RVB at high energies where the spin correlations, according to the authors, are only nearest neighbours, and a gapless nature of a spin liquid at low energy, that should have long range correlations (not seen experimentally as far as I understood), and a quite large value of the spin susceptibility.

The theoretical consistency of the claim is therefore lacking, but for this reason the paper may be very interesting and deserves to be published.

Reviewer #2 (Remarks to the Author):

This is my second review on this manuscript which focuses on the nature of the magnetic correlations in the rare-earth based triangular antiferromagnet YbMgGaO₄. I very much appreciate the efforts of the authors to clarify their work and to respond to my and other referees comments. After studying the two other referee reports and the detailed response by the authors, I find this manuscript still suffers from a major flaw, namely an erroneous interpretation of the data. Given the high quality of the experimental work, I am willing to give the authors' a last chance to modify their manuscript to reach the standard expected for Nature Communications.

Compared to previously published neutron work on this material, the ambition of the authors' is to "propose a different interpretation of these high-energy excitations and also endeavor to probe YbMgGaO₄ at lower energies". I elaborate below on flaws related to both high- and low-energies data interpretation.

-- High-energy excitations: the authors' interpretation relies on the correspondence between a nearest-neighbor resonating valence-bond (RVB) model and their data above 0.5 meV. As pointed out by Referee 1, this correspondence may merely indicate that spins in YbMgGaO₄ are at the vertices of a triangular lattice. High-energy excitations only reflect short distance correlations and will always somewhat look like a nearest-neighbor RVB state. One way the authors can convince themselves of this fact is to calculate the spin-wave spectrum for the 120 degree structure (long-range ordered state ubiquitous in Heisenberg triangular-lattice antiferromagnets) and integrate excitations along energy from half the bandwidth to the top of the band. It will look like an apparent nearest-neighbor RVB state although the ground-state is actually long-range ordered! The flaw in the authors' reasoning is, as I pointed out in my first report, that low-energy excitations MUST be included to calculate the equal-time correlator. The equal-time correlator (integration from zero energy to the top of the band) follows the zeroth-moment sum-rule and is thus an adequate quantity to compare different diffuse scattering models. I understand a similar modeling is done for Herbersmithite (Han et al., Nature 2012) but in the latter case the bandwidth is 10+meV and the integration starts at 1 meV what is somewhat below the FWHM energy resolution of the spectrometer, thus long-range correlations are somewhat captured. In the present case, the bandwidth is 1.5-2 meV and the authors' start their integration at 0.5 meV, what is several times the energy resolution, and thus longer range correlations are not captured.

-- Low-energy excitations: the authors write: "We, therefore, expect that below 0.13 meV the INS intensity at 0.1 K falls below that at 35 K. ... the intensity difference $I(0.1K) - I(35K)$ at zero magnetic field changes sign and becomes negative at energy transfer below J_0 [0.13 meV]". This statement is contradicted by the authors own Fig. R8 panel (c) which shows a positive $I(0.1K) - I(35K)$ for energies $|E| < 0.1$ meV for wave-vectors near the side of the Brillouin zone boundary (M-point). I believe this extra positive intensity is of the utmost importance for the above

consideration of the equal-time correlator. It would shift the overall intensity from the K-point to the M-point! I understand that the sample environment background may have changed between 0.1 K and 35 K measurements (mixture out, introduction of exchange gas) but I believe the signal in R8(c) is from the sample and is magnetic. As the main part of the paper presents energy-dependence plots integrated over the entire Brillouin zone, this extra low-energy signal is missed. Constant-momentum cuts in selected patches of the Brillouin zone are needed to reveal it.

In conclusion, the two problems discussed above are serious and unfortunately, they prevent me from recommending this paper for publication in Nature Communications at this time.

Reviewer #3 (Remarks to the Author):

I've read through the authors responses to my comments as well as the new version of the manuscript. I had 5 specific comments, and I believe the authors responded to all of them quite well. They also have given detailed responses to the comments of the other two referees.

While I continue to feel that disorder likely plays a very significant role in the physics of the ground state of this material, this subject is now discussed in the manuscript, and the manuscript itself is itself much improved. This will probably not be the "final word" on the subject, but it is an interesting contribution on a very topical material. I therefore now recommend acceptance.

Reviewer #1 (Remarks to the Author):

The authors have replied in a satisfactory manner most of my concerns. Before publication I recommend only to elaborate on the meaning of an RVB at high energies where the spin correlations, according to the authors, are only nearest neighbours, and a gapless nature of a spin liquid at low energy, that should have long range correlations (not seen experimentally as far as I understood), and a quite large value of the spin susceptibility. The theoretical consistency of the claim is therefore lacking, but for this reason the paper may be very interesting and deserves to be published.

Reply: We thank Reviewer #1 for recommending the publication of our manuscript. We gladly make the revisions proposed by the Reviewer.

Reviewer #2 (Remarks to the Author):

This is my second review on this manuscript which focuses on the nature of the magnetic correlations in the rare-earth based triangular antiferromagnet YbMgGaO₄. I very much appreciate the efforts of the authors to clarify their work and to respond to my and other referees comments. After studying the two other referee reports and the detailed response by the authors, I find this manuscript still suffers from a major flaw, namely an erroneous interpretation of the data. Given the high quality of the experimental work, I am willing to give the authors' a last chance to modify their manuscript to reach the standard expected for Nature Communications.

Compared to previously published neutron work on this material, the ambition of the authors' is to "propose a different interpretation of these high-energy excitations and also endeavor to probe YbMgGaO₄ at lower energies". I elaborate below on flaws related to both high- and low-energies data interpretation.

Comment-1: -- *High-energy excitations: the authors' interpretation relies on the correspondence between a nearest-neighbor resonating valence-bond (RVB) model and their data above 0.5 meV. As pointed out by Referee 1, this correspondence may merely indicate that spins in YbMgGaO₄ are at the vertices of a triangular lattice. High-energy excitations only reflect short distance correlations and will always somewhat look like a nearest-neighbor RVB state. One way the authors can convince themselves of this fact is to calculate the spin-wave spectrum for the 120 degree structure (long-range ordered state ubiquitous in Heisenberg triangular-lattice antiferromagnets) and integrate excitations along energy from half the bandwidth to the top of the band. It will look like an apparent nearest-neighbor RVB state although the ground-state is actually long-range ordered! The flaw in the authors' reasoning is, as I pointed out in my first report, that low-energy excitations MUST be included to calculate the equal-time correlator. The equal-time correlator (integration from zero energy to the top of the band) follows the zeroth-moment sum-rule and is thus an adequate quantity to compare different diffuse scattering models. I understand a similar modeling is done for Herbersmithite (Han et al., Nature 2012) but in the latter case the bandwidth is 10+meV and the integration starts at 1 meV what is somewhat below the FWHM energy resolution of the spectrometer, thus long-range correlations are somewhat captured. In the present case, the bandwidth is 1.5-2 meV and the authors' start their integration at 0.5 meV, what is several times the energy resolution, and thus longer range correlations are not captured.*

Reply: We thank Reviewer #2 for providing the possible explanation. However, we disagree with the Reviewer for the following reasons:

1. In herbersmithite, the minimum integration energy, $E_{\min} = 1$ meV, was reported to be *well above* the energy resolutions ($1/2\sigma$), 0.21 ($E_{\min}/\sigma = 2.4$) and 0.08 meV ($E_{\min}/\sigma = 6.3$) [half-width at half-maximum, $1/2\sigma$, please see METHODS SUMMARY in Nature 492, 406-410 (2012)]. In our case, we use a similar cutoff ($E_{\min}/\sigma = 0.5/0.16 = 3.1$) to exclude the elastic signal completely (see Fig. S14 in the Supplementary Information).

2. We followed the suggestion by the Reviewer and calculated spin-wave excitations for the hypothetical 120 degree structure [Phys. Rev. B 94, 035107 (2016)]. They do not look like "an apparent nearest-neighbor RVB state".

First, we use experimental parametrization of the spin Hamiltonian of YbMgGaO_4 ($J_{zz} = 0.085$ meV, $J_{\pm} = 0.078$ meV, $J_{\pm\pm} = 0.013$ meV and $J_{z\pm} = 0.003$ meV) [Phys. Rev. Lett. 115, 167203 (2015)], which on the classical level leads to the 120-deg ordered state. The INS spectrum calculated on the basis of the linear spin-wave theory [J. Phys. Condens. Matter 27, 166002 (2015)] (Fig. R1) is broadened with the experimental energy resolution of $\sigma = 0.16$ meV (Fig. R2). Note that the bandwidth of the spin-wave spectrum is only ~ 0.4 meV, much lower than in the experiment.

We integrate the spin-wave spectrum from half the bandwidth (~ 0.2 meV) to energies well above the top of the band (~ 0.5 meV), as proposed by the Reviewer. Magenta curves in Fig. R3 show that the resulting \mathbf{Q} -dependence is essentially inconsistent with that of our experimental data. For the sake of completeness, we also provide \mathbf{Q} -dependences obtained by integrating the spin-wave spectra starting from 0, 0.12, 0.3, 0.34 and 0.4 meV. None of them is even qualitatively consistent with our experimental data (see Fig. R3).

We also performed a spin-wave calculation for the 120-deg state of the simple nearest-neighbor Heisenberg Hamiltonian on the triangular lattice (Fig. R4-R6), arriving at very similar results. Even at high energies, the 120-deg state remains perfectly distinguishable from the RVB state, because these two states are fundamentally different. This essentially refutes the Reviewer's statement that any state on the triangular lattice produces same \mathbf{Q} -dependence of high-energy excitations. Neither we see any physical reason for that.

Spin-wave calculations for the 120-deg ordered state and the above analysis are included in the Supplemental Material.

Figure R1. Calculated spin-wave INS excitations [Rev. Sci. Instrum. 84, 083906 (2013)] along $[-1, K+0.5, 0]$, $[H, 0, 0]$, and $[-H, 0.5H, 0]$ for the hypothetical 120 degree magnetically ordered structure (see the inset for the magnetic unit cell) based on the experimental parametrization of the spin Hamiltonian of YbMgGaO_4 [Phys. Rev. Lett. 115, 167203 (2015)]. Here, $f(\mathbf{Q})$ is the magnetic form factor of Yb^{3+} .

Figure R2. Calculated spin-wave excitations (Fig. R1) with the Gaussian broadening, FWHM = 0.16 meV.

Figure R3. Wave-vector dependence of the INS intensity along $[-1, K+0.5, 0]$, $[H, 0, 0]$, and $[-H, 0.5H, 0]$, with blue lines representing nearest-neighbor RVB calculations, $a|F(\mathbf{Q})|^2 + b$. The spin-wave spectra based on the experimental parametrization of the spin Hamiltonian are integrated from 0, 0.12, 0.2, 0.3, 0.34, and 0.4 meV (all up to 0.5 meV) yielding the black, wine, magenta, red, green and violet curves, respectively. Above 0.3 meV, only minor changes in the \mathbf{Q} -dependence are observed showing that even at very high energies the \mathbf{Q} -dependence of the RVB state is not reproduced. The magnetic Bragg peak $(-2/3, 1/3, 0)$, which is naturally present in the spin-wave spectra, reflects long-range order in the 120-deg state and disappears at high energies.

Figure R4. Calculated spin-wave INS excitations along $[-1, K+0.5, 0]$, $[H, 0, 0]$, and $[-H, 0.5H, 0]$ for the 120 degree magnetically ordered ground state (see the inset for the magnetic unit cell) based on the nearest-neighbor spin-1/2 antiferromagnetic Heisenberg Hamiltonian on the triangular-lattice ($J > 0$).

Figure R5. Calculated spin-wave excitations (Fig. R4) with the Gaussian broadening, FWHM = 2J.

Figure R6. Wave-vector dependence of the INS intensity along $[-1, K+0.5, 0]$, $[H, 0, 0]$, and $[-H, 0.5H, 0]$, with blue lines representing nearest-neighbor RVB calculations, $a|F(\mathbf{Q})|^2 + b$. The spin-wave spectra for the nearest-neighbor Heisenberg Hamiltonian are integrated from 0J and 3.4J (both up to 4J) yielding the black and pink curves, respectively. The violet curves show the integration result for spin-wave excitations calculated for the YbMgGaO₄ Hamiltonian (the violet curves in Fig. R3).

Comment-2: -- Low-energy excitations: the authors write: "We, therefore, expect that below 0.13 meV the INS intensity at 0.1 K falls below that at 35 K. ... the intensity difference $I(0.1K) - I(35K)$ at zero magnetic field changes sign and becomes negative at energy transfer below J_0 [0.13 meV]". This statement is contradicted by the authors own Fig. R8 panel (c) which shows a positive $I(0.1K) - I(35K)$ for energies $|E| < 0.1$ meV for wave-vectors near the side of the Brillouin zone boundary (M-point). I believe this extra positive intensity is of the utmost importance for the above consideration of the equal-time correlator. It would shift the overall intensity from the K-point to the M-point! I understand that the sample environment background may have changed between 0.1 K and 35 K measurements (mixture out, introduction of exchange gas) but I believe the signal in R8(c) is from the sample and is magnetic. As the main part of the paper presents energy-dependence plots integrated over the entire Brillouin zone, this extra low-energy signal is missed. Constant-momentum cuts in selected patches of the Brillouin zone are needed to reveal it.

In conclusion, the two problems discussed above are serious and unfortunately, they prevent me from recommending this paper for publication in Nature Communications at this time.

Reply: We thank the Reviewer. We have checked our data carefully, and there is no such a contradiction. Below 0.13 meV, $I(0.1\text{ K})$ is indeed lower than $I(35\text{ K})$ when the intensity is integrated over *all* measured wave-vector (\mathbf{Q}) space, while the figure mentioned by the Reviewer (Fig. R7 c in the current version) is a \mathbf{Q} -dependent spectrum integrated over the energy $|E| \leq 0.1$ meV. The positive values of $I(0.1\text{ K}) - I(35\text{ K})$ appear only in a part of the measured \mathbf{Q} -space along with negative values in other regions of the \mathbf{Q} -space. These negative values remain dominant down to ~ 0.02 meV, whereas at even lower energies, $I(0.1\text{ K}) - I(35\text{ K})$, integrated over *all* measured \mathbf{Q} space, becomes positive again (see Fig. 3d in the main text).

Figure R7. The panel (a)-(c) of Fig. R8 from our first response file, or the panel (a)-(c) of Fig. S18 in the Supplementary Information.

We concur with the Reviewer that positive values of $I(0.1\text{ K}) - I(35\text{ K})$ for $|E| \leq 0.1$ meV around the M-point may be intrinsic, in agreement with [Nat. Phys. 13, 117-122 (2017)], where maxima of diffuse scattering at the M-point were observed. This in fact leads to a coherent picture of YbMgGaO_4 . At high energies, nearest-neighbor correlations dominate, and the intensity maximum is located at the K-point. At low energies, the intensity maximum shifts to the M-point reflecting the fact that long-range correlations become relevant. We demonstrate that these two regimes are well separated in energy. They seem to be demarcated by a gap, which we infer from negative values of $I(0.1\text{ K}) - I(35\text{ K})$ [integrated over the whole \mathbf{Q} -range] between ~ 0.02 meV and ~ 0.13 meV. Therefore, the broad excitation continuum above $\sim J_0$, previously described in terms of a spinon Fermi surface, is in fact due to nearest-neighbor correlations, presumably of RVB type, whereas at very low energies a different type of physics emerges.

We revised the discussion part of the manuscript and emphasized these aspects of our interpretation. We also refer to other recent literature, including the newly published preprints that likewise argue against the presence of the spinon Fermi surface in YbMgGaO_4 .

Reviewer #3 (Remarks to the Author):

I've read through the authors responses to my comments as well as the new version of the manuscript. I had 5 specific comments, and I believe the authors responded to all of them quite well. They also have given detailed responses to the comments of the other two referees.

While I continue to feel that disorder likely plays a very significant role in the physics of the ground state of this material, this subject is now discussed in the manuscript, and the manuscript itself is itself much improved. This will probably not be the "final word" on the subject, but it is an interesting contribution on a very topical material. I therefore now recommend acceptance.

Reply: We thank the reviewer for recommending the acceptance.

REVIEWERS' COMMENTS:

Reviewer #2 (Remarks to the Author):

The authors have greatly improve their manuscript through the referring process. I would like to thank for their thorough consideration of comments and concerns. The manuscript is now suitable for Nature Communications and should be published expediently.

RESPONSE TO REFEREES:

Reviewer #2:

The authors have greatly improve their manuscript through the referring process. I would like to thank for their thorough consideration of comments and concerns. The manuscript is now suitable for Nature Communications and should be published expediently.

Reply: We are delighted that Reviewer #2 recommended the publication of our manuscript. We thank the reviewer very much.